# A Maastrichtian insect assemblage from Patagonia sheds light on arthropod diversity previous to the K/Pg event

Ezequiel I. Vera[1], Mateo D. Monferran[2], Julieta Massaferro [3], Lara M. Sabater[2], Oscar F. Gallego[2], Valeria S. Perez Loinaze[1], Damián Moyano-Paz[4], Federico L. Agnolín[5,6], Makoto Manabe [7], Takanobu Tsuhiji[8] & Fernando E. Novas [5✉]

Insect faunas from the latest Cretaceous are poorly known worldwide. Particularly, in the Southern Hemisphere, there is a gap regarding insect assemblages in the Campanian-Maastrichtian interval. Here we present an insect assemblage from the Maastrichtian Chorrillo Formation, southern Argentina, represented by well-preserved and non-deformed, chitinous microscopic remains including head capsules, wings and scales. Identified clades include Chironomidae dipterans, Coelolepida lepidopterans, and Ephemeroptera. The assemblage taxonomically resembles those of Cenozoic age, rather than other Mesozoic assemblages, in being composed by diverse chironomids and lepidopterans. To the best of our knowledge, present discovery constitutes the first insect body fossils for the Maastrichtian in the Southern Hemisphere, thus filling the gap between well-known Early Cretaceous entomofaunas and those of Paleogene age. The presented evidence shows that modern clades of chironomids were already dominant and diversified by the end of the Cretaceous, in concert with the parallel radiation of aquatic angiosperms which became dominant in freshwater habitats. This exceptional finding encourages the active search of microscopic remains of fossil arthropods in other geological units, which could provide a unique way of enhancing our knowledge on the past diversity of the clade.

[1] División Paleobotánica, Museo Argentino de Ciencias Naturales "Bernardino Rivadavia" (MACN-CONICET), Av. Ángel Gallardo 470, C1405DJR Ciudad Autónoma de Buenos Aires, Argentina. [2] Centro de Ecología Aplicada del Litoral CONICET y Departamento de Biología, FaCENA-UNNE, Ruta Provincial Nº 5, s/n, Km 2,5, 3400 Corrientes, Argentina. [3] Programa de Estudios Aplicados a la Conservación de la Biodiversidad CENAC/APN, Fagnano 244, 8400 Bariloche, Argentina. [4] Centro de Investigaciones Geológicas (CIG, CONICET-UNLP), Diagonal 113 #275, B1904DPK La Plata, Buenos Aires, Argentina. [5] Laboratorio de Anatomía Comparada y Evolución de los Vertebrados (LACEV), Museo Argentino de Ciencias Naturales "Bernardino Rivadavia" (MACN-CONICET), Av. Ángel Gallardo 470, C1405DJR Ciudad Autónoma de Buenos Aires, Argentina. [6] Fundación de Historia Natural "Félix de Azara", Departamento de Ciencias Naturales y Antropología, Universidad Maimónides, Hidalgo 775, C1405BDB Ciudad Autónoma de Buenos Aires, Argentina. [7] Center for Collections, National Museum of Nature and Science, Tsukuba 305-0005, Tokio, Japan. [8] Department of Geology, National Museum of Nature and Science, 3-23-1 Hyakanin-cho, Shinjuku-lu, 1069-0073 Tokyo, Japan. ✉email: fernovas@yahoo.com.ar

The fossil record of insects known for the Late Cretaceous is scarcely known across the globe and, in particular, it is almost inexistent in the Campanian-Maastrichtian interval[1–3]. Information on Campanian-Maastrichtian insect diversity in South America consists on nests, pupal chambers, cocoons, and feeding activities on leaves and wood, recovered in central and northern Patagonia[4–9]. These occurrences suggest a complex community of terrestrial insects represented by different types of herbivores, pollen feeders, and predators/parasites of scavengers (wasps). However, body insect remains are still wanting. Further, there are no fossils belonging to aquatic insects from this time interval in South America (except for a mention, without descriptions or illustrations, of nepomorph waterbugs in the La Colonia Formation[10]), which are otherwise of very rare occurrence in the global record[11].

Here we present an insect assemblage from the lower Maastrichtian Chorrillo Formation, cropping out in Santa Cruz province, southern Argentina. Previous uppermost Cretaceous report from insects in the region consisted of coleopteran elytra imprints presumably from the lowermost levels of the Dorotea Formation in southern Chile[12,13]. The Chorrillo Formation has yielded a rich and variegated fossil content[14] including titanosaurids, hadrosaurids, ankylosaurids[15], non-avian theropods[16], enantiornithine birds[17,18], gondwanatherian and monotreme mammals[19–21], turtle, frogs, and fishes[14,17], terrestrial mollusks[14], and conifer woods, pollen and spores of different plant groups[14,17,22,23]. The entomofauna currently recognized is composed by members of Chironomidae, Ephemeroptera, and Lepidoptera. Insect diversity may be greater, as suggested by the presence of abundant chitinous remains belonging to taxonomically unidentified arthropods. To the best of our knowledge, these constitute the first insect assemblage from the Maastrichtian of southern South America, offering information about the diversity of water dwelling insects previous to the Cretaceous–Paleocene massive extinction event.

Recovered arthropod remains consist of well-preserved cephalic capsules of aquatic larvae of chironomids, isolated scales and fragments of larval exuviae of lepidopterans, and an ephemeropteran larval head, as well as diverse fragmentary remains of unknown arthropod affinities. They are within the size range of palynomorphs (c. 10–250 µm) and were identified in palynological slides. The discovery of these chitinous remains was fortuitous, after preparing fragments of rocks with the usual techniques employed for palynology (see Methods). It is worth noting that, for historical reasons, palynologists working on pre-Quaternary material infrequently focus on the study of arthropod remains associated with palynomorphs[24], and they are rarely included in the interpretation of the data[25–28]. The not-deformed, chitinous remains from both aquatic and terrestrial insects documented in palynological samples from the Chorrillo beds, constitute an exceptional preservation for Upper Cretaceous fossils, resembling those frequently found in Quaternary samples[29].

## Results

Arthropod materials are represented by approximately 30 specimens, and include representatives of the Chironomidae, Ephemeroptera, and Lepidoptera, as well as remains of unknown affinities.

## Systematic Paleontology.

Phylum Arthropoda Gravenhorst, 1843
Subphylum Hexapoda Latreille, 1825
Class Insecta Linnaeus, 1758
Order Diptera Linnaeus, 1758
Suborder Nematocera Latreille, 1825
Family Chironomidae Macquart, 1838

*Subfamily Orthocladiinae Lenz, 1921*
Morphotype 1: *Referred specimens.* MPM-Pal 21835-24: M31/0, MPM-Pal 21835-3: P46/4.

*Description.* The specimen 1 (MPM-Pal 21835-24: M31/0, Fig. 1a) corresponds to a head capsule (135 µm long and 77 µm high) showing a (partial) mentum (at least 43 µm long and 9 µm high) with 5 lateral teeth and one mandible (41 µm long and 25 µm wide at its base) showing an apical short tooth and 3–4 inner teeth similar in shape.

Specimen 2 (MPM-Pal 21835-3: P46/4, Fig. 1b) preserves a complete mentum (47 µm long and 10 µm high) with double median teeth and 5 pairs of lateral teeth, and a single mandible with an apical short tooth and 4 inner teeth. Both morphotypes exhibit sclerotized narrow ventromental plates, extending through the outermost lateral tooth.

The shape and position of the ventromental plates support these morphotypes unequivocally belong to the Subfamily Orthocladiinae (Fig. 1d). The double median teeth and the mandible resemble the living genera *Parapsectrocladius* Cranston and a *Botryocladius* Cranston of the Orthocladiinae[30,31], but preservation of the available remains are not enough for their correct identification.

Morphotype 2: *Referred specimens.* MPM-Pal 21835-25: V26/3..
*Description.* The specimen (Fig. 1c) consists of a mentum (38 µm wide and 22 µm high) detached from the head capsule. Double median teeth and 5 pairs of lateral teeth, median teeth slightly taller and wider than 1st lateral; 2nd lateral tooth only slightly smaller than 1st lateral tooth. All teeth are pointed, taller than wide. The shape of the mentum in joint with the inconspicuous ventromental plate, are features typical of an orthoclad (Fig. 1d); however, the mentum lacks similarities with any known living chironomid, thus it is interpreted as Orthocladiinae *incertae sedis*.

*Subfamily Diamesinae Pagast, 1947.*
Tribe Heptagyini Brundin, 1966

Morphotype 3: *Referred specimens.* MPM-Pal 21835-9:E34/4, MPM-Pal 21835-19: V34/4.

*Description.* This morphotype (Fig. 1e) consists of isolated menti (136 µm wide and 14 µm high) showing double median tooth and 7–11 lateral teeth. The arrangement of mental teeth, however, gives an appearance of 3–6 median teeth. The presence of at least nine lateral teeth allows its placement in the Diamesinae, a clade of chironomids often bearing more than seven lateral teeth on the mentum (Fig. 1f). The morphotype is similar to the living genus *Paraheptagyia* Brundin, although in the later one there is not differentiation of central teeth, suggesting it may represent an extinct taxon.

*Subfamily Tanypodinae sensu Cranston et al., 1983*
Morphotype 4: *Referred specimens.* MPM-Pal 21835-2:Y40/3
*Description.* The specimen (Fig. 1g) consists of a ligula (12 µm wide and 15 µm high) showing five teeth, the two outer teeth dark brown and the three central pale brown teeth. The ligula is a common character of all the genera within the Subfamily Tanypodinae (Fig. 1h).

*Subfamily indeterminate*
Morphotype 5: *Referred specimens.* MPM-Pal 21835-9: Q58/3.

*Description.* The specimen (Fig. 1i) shows two overlapped menti and a mandible in the same slide. The left side of the mentum is approximately 31 µm long and 7 µm high, and is similar to morphotypes 1 and 2 of Subfam. Orthocladiinae, whereas the right mentum (18 µm long and 7 µm high) looks similar to morphotype 4 (Diamesinae). In addition, the mandible

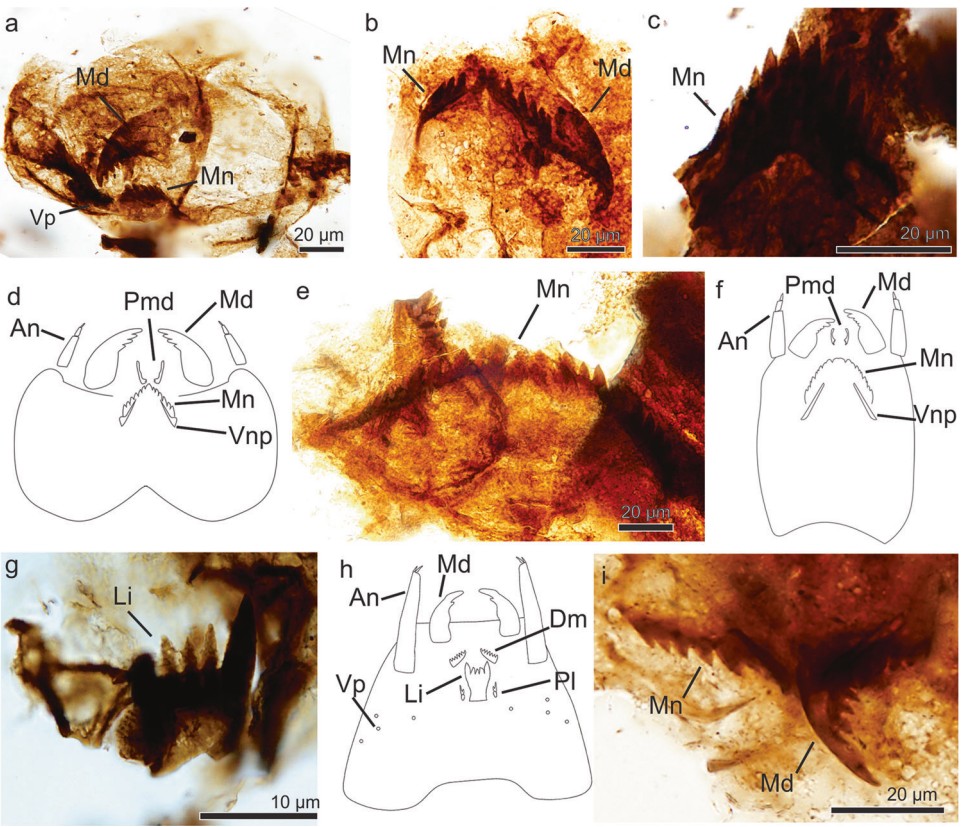

**Fig. 1 Arthropod fossil remains from the Chorrillo Formation referred to Chironomidae. a–d** Orthocladiinae. **a**, **b**, Morphotype 1; **a** MPM-Pal 21835-24: M31/0; **b** MPM-Pal 21835-3: P46/4; **c** Morphotype 2, MPM-Pal 21835-25: V26/3; **d** Schematic representation of an Orthocladiinae head capsule; **e**, **f** Diamesinae, **e**, Morphotype 3, MPM-Pal 21835-19: V34/4; **f**, Schematic representation of a Diamesinae head capsule; **g**, **h** Tanypodinae, **g** Morphotype 4, MPM-Pal 21835-2:Y40/3; **h** Schematic representation of a Tanypodinae head capsule; Morphotype 5, MPM-Pal 21835-9: Q58/3. An antennae, Li ligulae, Md mandible, Mnt mentum, Plg paraligulae, Prm premandible, Vnp Ventromental plate, Vp ventromental pores. **d**, **f**, and **h** based on personal photo library of J. Massaferro.

(26 μm long and 11 μm wide at its base) does not seem to correspond to any of the aforementioned morphotypes, but it resembles the mandible of *Cricotopus* spp[32].

### Order Ephemeroptera Hyatt and Arms, 1890
Ephemeroptera indet: *Referred specimens*. MPM-Pal 21835-9:W37/1

*Description*. The specimen (Fig. 2a) corresponds to a small (340 μm long and 260 μm high) but heavily sclerotized head. A pair of small subcircular compound eyes is located near the posterolateral corners of the head, bearing seven rows of lenses (ommatidia). The eyes are anteriorly delimited by prominent ocelar tubercles. The head surface is moderately densely covered with hairlike setae. Patches of short, hairlike setae are present below and laterally to the compound eyes. The mouthparts are incomplete and not well preserved, the labrum and pharynx are obscured by other structures. The head is prognathous and shows the presence of a region with strong sclerotization and acuminate ends that suggest that a pair of mandibles were well-differentiated. There is no evidence of mandibular tusk.

*Remarks*. The presence of sclerotized mandibles which are not expanded and lack prominent teeth, and transversely expanded head with eyes located at the posterolateral corners, indicate that it corresponds to a Ephemeroptera naiad[33–35] (Fig. 2b). Further, differs from the closely related odonates in that the latter lack sclerotized mandibles[36]. Presence of ocelar tubercle delimiting the anterior margin of the eye and located at the corner of the head occurs in the extant families Caenidae and Ephemerellidae[35].

The fossil record shows that mayflies were abundant and extremely numerous regionally during the Cretaceous, being reported from Asia, North America, and the early Cretaceous of Africa and Australia[36–42]. In South America they are restricted to the Aptian–Albian Crato Formation of Brazil, where more than 17 species have been described[42], and a single finding in the Lower-Upper Cretaceous of Colombia[43].

### Order Lepidoptera Linnaeus, 1758
Lepidoptera indet: *Referred specimens*. MPM-Pal 21835-10:X30/2, MPM-Pal 21835-7:S27/2

*Description*. Two chitinous remains are tentatively identified as belonging to lepidopteran larval exuvial remains. One of these is at least 227 μm.long and 74 μm wide (Fig. 3a), and shows at least four pairs of 23–30 μm long thoracic legs. The other one (Fig. 3c) probably corresponds to the last sommite of a larval remain, at least 253 μm long bearing the anal proleg with semicircular shaped uniordinal crochets, 31.5 μm.in diameter.

*Remarks*. Presence of thoracic legs and prolegs, the later ones exhibiting uniordinal crochets, could indicate that the studied elements are probably remains of a lepidopteran larval cuticle. The presence of a single series of more or less uniform (uniordinal) crochets observed in the specimens, is the plesiomorphic condition for lepidopterans[44] (Fig. 3b). However, there are several insect groups that have abdominal appendices known as prolegs[45,46], and there is a considerable diversity/variation in their number, position, and shape[47], and thus, the specimens here reported are identified as Lepidoptera indet.

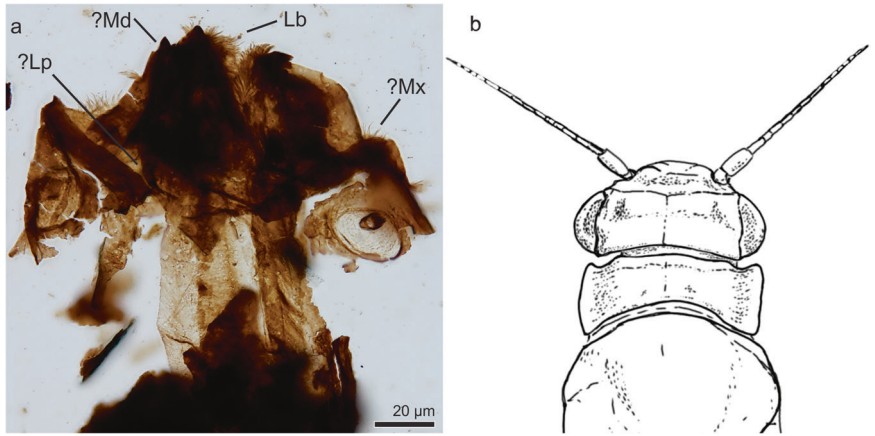

**Fig. 2 Arthropod fossil remains from the Chorrillo Formation referred to Ephemeroptera. a** Ephemeroptera head MPM-Pal 21835-9:W37/1; **b** lineal drawing of a living caenid head for comparisons (based on Caenidae nymph.jpg by Dmccabe, 2017 CC BY SCA 4.0 - REF [85]). Md mandible, Mx maxilla, Lb labrum, Lp labial palpus.

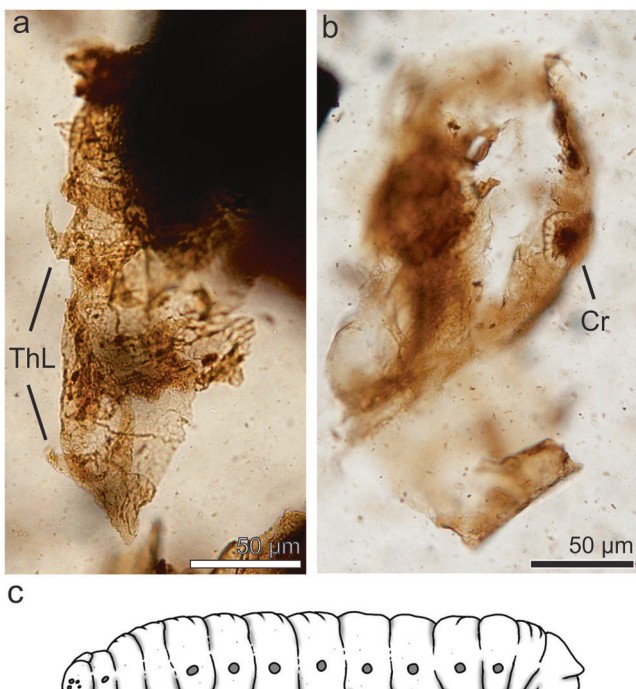

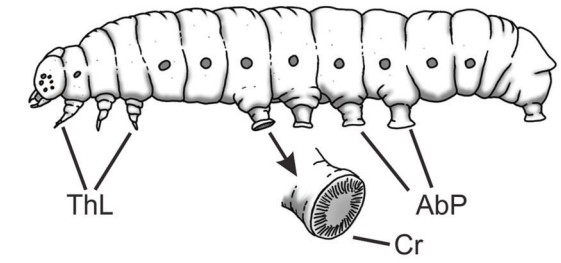

**Fig. 3 Arthropod fossil remains from the Chorrillo Formation referred to Lepidoptera. a** Larval exuvial fragment showing thoracic legs, MPM-Pal 21835-7:S27/2; **b** Probable last sommite larval exuvial fragment showing crochet-bearing prolegs, MPM-Pal 21835-10:X30/2; **c** Lineal drawing of an extant lepidopteran larva for comparison (based on Lukas3, public domain. Gasienica_motyla_budowa.svg; REF [86]), and on Moreira et al., 2019 (Fig 33 CC by-SA 4.0; REF [87]). AbP abdominal prolegs, Thl thoracic legs, Cr crochets.

*Suborder Glossata Fabricius, 1775.*
Clade Coelolepida Nielsen and Kristensen, 1996

Coelolepida indet: *Referred specimens.* MPM-Pal 21835-3:W34/1, MPM-Pal 21835-23:W55/0, MPM-Pal 21835-22:R52/2,MPM-Pal 21835-20:Z47/3. MPM-Pal 21835-22: G55/0; MPM-Pal 2183-21: E42/3

*Description.* At least four morphological types of scales and scale fragments were recovered. They mainly differ in size, general shape, apical margin shape, and presence of perforations (Fig. 4). All the specimens bear longitudinal ridges, cross ridges, and small inter-ridge perforations. In addition, they exhibit a clear pedicel below their basal margin, and the longitudinal ridges seemingly do not extend beyond the apical margin. Broken scales indicate that they are hollow.

Type 1 scale (specimen MPM-Pal 21835-3:W34/1; Fig. 4a, b) is 125 μm long and 42 μm wide, and shows longitudinal ridges (Fig. 4b), serrated apical margin, and rounded basal margin.

Type 2 (specimen MPM-Pal 21835-23:W55/0; Fig. 4c) is a fragmentary scale 147 μm long and 26 μm wide, similar to Type 1, but with a serrated apical margin having finger-like projections.

Type 3 scale (specimen MPM-Pal 21835-22:R52/2; Fig. 4d) is 178 μm long and approximately 48 μm wide, with a rounded apical margin, and a large pedicle.

Type 4 (specimens MPM-Pal 21835-20:Z47/3; MPM-Pal 21835-22: G55/0; MPM-Pal 2183-21: E42/3; Fig. 4e) is long and dark in a central zone, reaching 174 μm long and 30 μm wide, and bears a slightly serrated apical margin.

*Remarks.* The scales described above are similar to the ones present in the lepidopteran clade Coelolepida[48] due to the presence of serrated apical margins, dense cross-ridges, hollow structure, and perforations in the inter-ridge areas of the upper lamina, characteristic of the vast majority of extant Lepidoptera[27,48]. Rhaetian–Hettangian lepidopterans and non-glossatan moths differ in having rounded scales, with a solid inner structure[47]. Trichopterans were recently described for Cretaceous deposits[49], but they exhibit scales lacking serrated apical margins, and cross-ridges[47], thus differing from lepidopteran scales recorded in the Chorrillo Formation.

In spite that the earliest lepidopteran body fossil dates back to the Early Jurassic of England[50], lepidopteran wing scales were

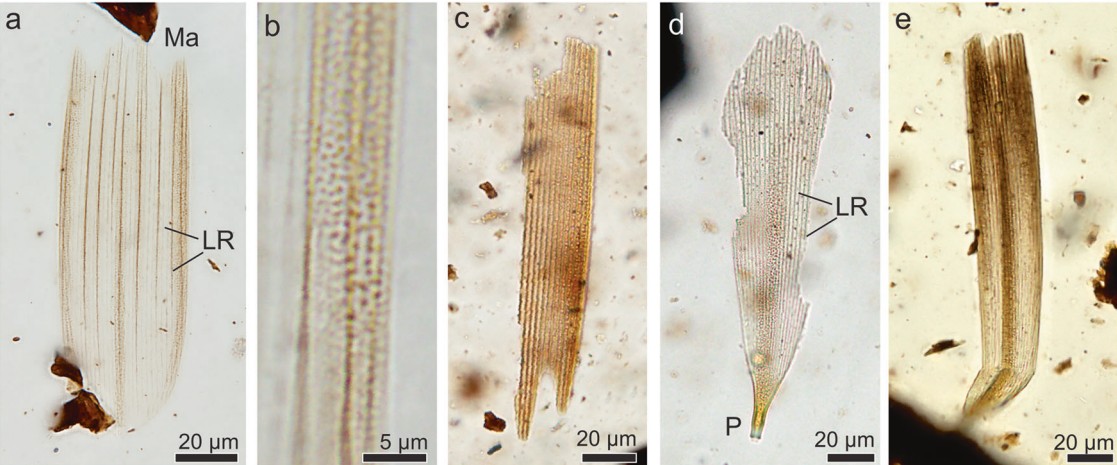

**Fig. 4 Arthropod fossil remains from the Chorrillo Formation referred to Lepidoptera. a** Type 1 scale, MPM-Pal 21835-3:W34/1; **b** Detail of Type 1 scale showing small inter-ridge perforations; **c** Type 2 scale, MPM-Pal 21835-23:W55/0; **d** Type 3 scale, MPM-Pal 21835-22:R52/2; **e** Type 4 scale, MPM-Pal 21835-20:Z47/3. Ma serrated apical margin, LR longitudinal ridges, P pedicle.

recorded from Triassic–Jurassic boundary sediments from northern Germany[47,51].

Lepidoptera are scarce in fossil records from South America, with representatives of the microlepidopteran grade occurring in the Crato Formation[52–54].

Indeterminate arthropod remains: *Description.* The Chorrillo insect assemblage also includes small arthropod remains of uncertain systematic position (Figs. 5 and 6). They hold irregularly distributed pores, setae, compound eyes, and spines. The recorded arthropod cuticles have different shapes and sizes and, in most cases, they range from 100 to 150 μm wide. Among them, there are fragmentary remains of compound eyes (Fig. 5d). However, none of these cuticles seem to be associated with any other diagnostic parts of other arthropods identified in the Chorrillo assemblage.

## Discussion
Currently known insect record of the Chorrillo Formation includes taxa representative of the Chironomidae, Ephemeroptera, and Lepidoptera, as well as arthropod remains of unknown affinities.

The presence of larval (Chironomidae) and naiad (Ephemeroptera) remains unambiguously indicates a freshwater environment for the Megafloral level 1, in agreement with the paleobotanical content and sedimentological features[17,22]. The larval instars of living chironomids inhabit an enormous variety of aquatic and semi-terrestrial environments, from moist soils to pools in tree holes, and from low-oxygen lake sediments to fast-flowing mountain streams[55]. The presence of Orthocladiinae and Diamesinae in the Chorrillo Formation suggest that Megafloral level 1 accumulated under low productive and vegetated, shallow waters. Even though, Diamesinae is usually a typical inhabitant of running waters, its presence in a lacustrine environment is not surprising as it could have been transported and later deposited in the lacustrine environment by erosion and/or precipitation runoff. Indeed, these insects were found together with aquatic plant remains in the Megafloral level 1 which was described as a stratum of swampy, low-energy environment located in the distal portion of low gradient meandering river floodplain (see Geological and paleoenvironmental contexts).

Morphotypes of both subfamilies are characterized by having mixed functional feeding strategies, including detritivores, herbivores, and predators. Chironomid larvae represent one of the

main proteinaceous bases of freshwater food chains of post-Mesozoic ecosystems, mainly for fish and other aquatic predators[56]. The Chorrillo Formation assemblage is taxonomically congruent with those of Cenozoic ecosystems because chironomids are numerically dominant[57]. In contrast, less than 10% of the specimens collected in Cretaceous amber inclusions correspond to Chironomidae[36,58]. While chironomids (in particular larval stages) are typically rare or absent in amber deposits, as chances of being caught in the resinous secretions of the trees are scarcer in the aquatic environment, its absence in Cretaceous lithic sediments is also noteworthy and, as far as we are aware, only a partial wing of a possible chironomid has been so far reported from the Early Cretaceous of Brazil[59].

Although available chironomid specimens collected in the Chorrillo Formation consist of fragmentary mouthparts, their morphological features show that they can be comfortably referred to extant tribes. This supports previous interpretations[60–62] that Chironomidae were morphologically stable since the Early Cretaceous, and that diversification of most of their tribes occurred during the Cretaceous, mainly between 118 through 66 Mya[63]. Moreover, the evidence afforded here dismisses the hypothesis[57,64–66] interpreting that chironomid subclades did not survive the K/Pg boundary.

Some authors[62,64] concluded that primitive chironomids (e.g., Buchonomyiinae, Aenneinae) were replaced during the Cenozoic by assemblages dominated by modern chironomid clades such as Orthocladiinae and Chironominae. This ecological shift has been correlated with the emergence of angiosperm-dominated deciduous forests, which led to the enhanced transport of allochthonous organic matter into the water bodies. Evidence yielded by the Chorrillo Formation shows that modern clades of chironomids were already abundant and diversified by the end of the Cretaceous, in concert with the radiation, for the same period, of aquatic angiosperms which became dominant in freshwater habitats[10].

The presence of Lepidoptera in the assemblage is congruent with the evolutionary renewal described above for chironomids, although in this case the four different morphotypes of wing scales may not be a direct indicator of taxonomic richness, as variable morphology is recognized in the vesture of lepidopterans[67]. In spite of this variation, the distinctions among morphological types observed in scales described above, are higher than that known in extant lepidopterans[67]. As for example, Type 2 scale lacks pedicle, bears a low number of longitudinal

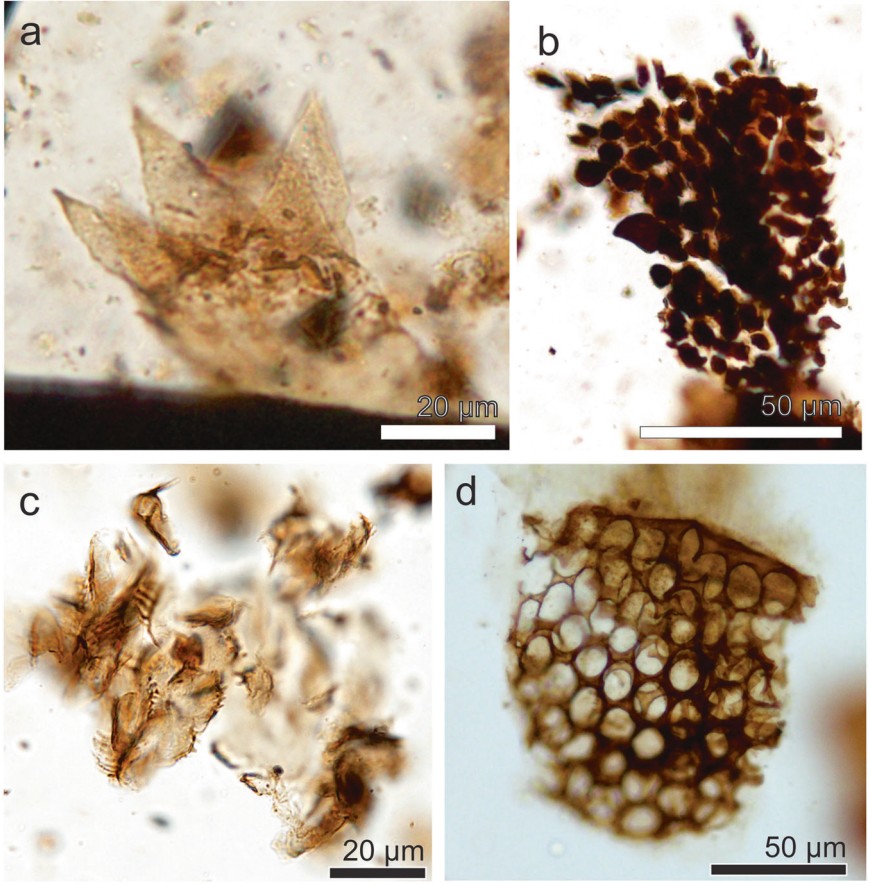

**Fig. 5 Fossil remains of indeterminate arthropods from the Chorrillo Formation. a** Terminal process with six spine-like projections, MPM-Pal 21835-7:G42/4; **b** Cuticule fragment with small process, MPM-Pal 21835-23:H55/0; **c** Cuticule fragment with well-conspicuous setae at the margins, MPM-Pal 21835-20:U27/0; **d** Fragment of compound eye, MPM-Pal 21835-4.

ridges, and its margin is decorated with digitiform serrations, being sharply different from Type 4, which has pedicle, the number of ridges is notably higher, a dark central area is present, and the distal margin is weakly serrated. These strong morphological differences are not found in the vestiture of a single extant lepidopteran species, making the referral of all scale types to a single biological species unlikely. In any case, the record of lepidopteran scales in this freshwater environment, along with exuvial remains also allied with the Lepidoptera, may indicate the presence of aquatic or semi-aquatic lepidopterans, conforming a small, taxonomically diverse, and scarcely known group[68]. This may also be indicative that lepidopterans may have acquired an outstanding role as pollinators and probably nectarivores by the latest Cretaceous. Such condition is reminiscent to Cenozoic insect associations, in which lepidopterans constitute one of the most frequent and diversified clades of phytophagous insects[36,69,70], contrasting with the Early Cretaceous times, when lepidopterans were numerically scarce[52–54,59].

The chironomid assemblage of the Chorrillo Formation resembles those of the Andean-Patagonian biogeographic region by the absence of Chironominae and the abundance and diversity of Orthocladiinae[71], and differs from those reported for Tropical lowlands or Neotropical Region[58]. These austral taxa generally inhabit cool, pristine environments, often upland to montane streams[61]. Noteworthy, some floristic elements recovered from the Chorrillo Formation, as well as paleosoil analysis, contrarily indicate temperate to warm conditions[23,72].

Chironomids recorded in the Chorrillo beds are anatomically close to some Australasian taxa, such as the Heptagyini diamesine *Paraheptagya* and the Orthocladiinae *Parapsectrocladius*, *Botryocladius,* and *Limnophyes*, currently present in the Andean and Australasian regions[30,31,61,73,74]. The strong tie between chironomids from Chorrillo beds and those of the Australasian Region is congruent with the evidence afforded by different fossil vertebrates: in the Chorrillo beds megaraptorid theropods are frequent, alongside with abundant elasmarian ornithischians[14,15,75], an assemblage resembling Cretaceous faunas of Australia. Recently, the monotreme *Patagorhynchus pascuali* was reported from the Chorrillo Formation, which exhibits close affinities with the Cenozoic ornithorhynchid *Obdurodon* from Australia[21], thus lending support to Transantarctic relationships with Australasia during Late Cretaceous times.

Summing up, the discovery of the Chorrillo Formation insect assemblage partially fills the gap between well-known Early Cretaceous entomofauna and those of Paleogene age. To our knowledge, only remains of insects of terrestrial and herbivorous habits were known from the Maastrichtian of Patagonia, lacking evidences of aquatic forms. The Chorrillo Formation insect assemblage resembles Cenozoic, rather than Mesozoic, insect assemblages in the abundance and diversity of chironomids, particularly of the Orthocladiinae/Diamesinae lineage, and in diversity of lepidopterans. This suggests that modern insect assemblages were already established in southern Patagonia for end of the Cretaceous Period.

## Methods

The fossil specimens reported in this publication were obtained from the upper Maastrichtian Chorrillo Formation after a field

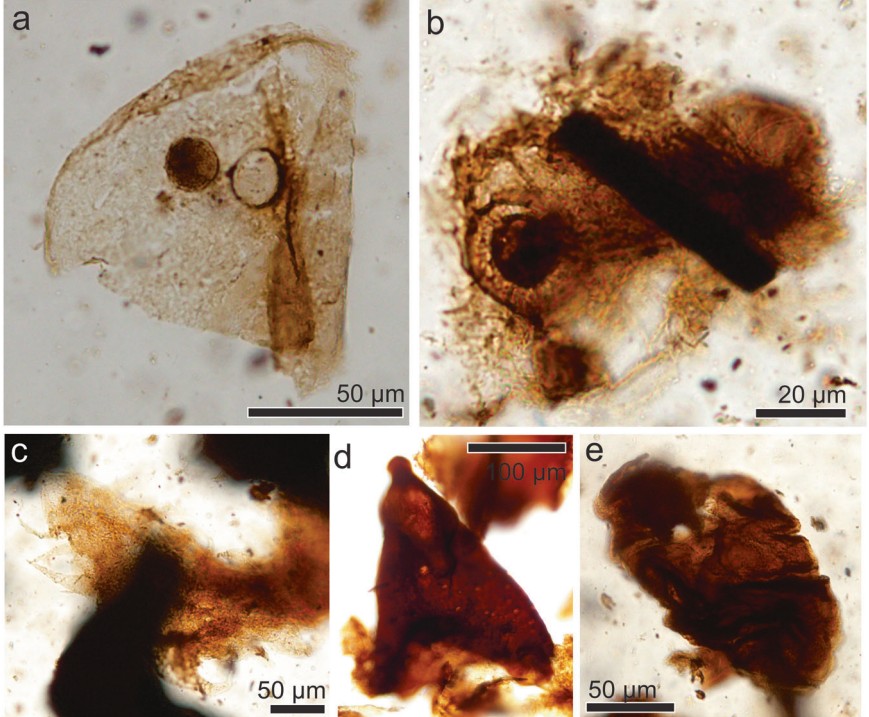

**Fig. 6 Fossil remains of indeterminate arthropods from the Chorrillo Formation. a** Folded cuticle fragment, MPM-Pal 21835-22:W55/2; **b** Cuticle fragment of arthropod affinity, MPM-Pal 21835-7:K21/2; **c** Cuticle fragment of possible arthropod exuvial origin, MPM-Pal 21835-23:L36/0; **d** Stiff conical structure, MPM-Pal 21835-4:R46/0; **e** Oval shape cuticle with wrinkles, MPM-Pal 21835-23:W46/0.

trip carried out near El Calafate city, Santa Cruz Province, Argentina, during March 2020. Exploration and fossil collecting was authorized by Secretaría de Estado de Cultura de la Provincia de Santa Cruz, Argentina. Studied specimens are deposited in the Museo Regional Padre Jesús Molina (Río Gallegos, Santa Cruz Province), under MPM-Pal 21835 (Palynological slides) acronyms.

**Geological and paleoenvironmental contexts**. The Chorrillo Formation is a syn-orogenic stratigraphic unit that accumulated during the early Maastrichtian in the Austral-Magallanes Basin during a continental phase of the basin[17,76]. The unit crops out on the south of the Lago Argentino, southwest of the Santa Cruz province, Argentina (Fig. 7A) and it overlies the coarse-grained braided fluvial deposits of the La Irene Formation[77,78] and is covered by the shallow marine deposits of the Calafate Formation[79] (Fig. 7B). The Chorrillo Formation is a~500 m thick sequence (Figs. 7C and 8) interpreted as accumulated in a low gradient fluvial environment dominated by thick fine-grained floodplain deposits which alternate with channel-shaped and lobe-shaped sandstone bodies interpreted and fluvial channels and crevasse splay facies[17]. The fine-grained deposits constitute more than 60% of the unit and show evidence of both soil development and waterlogging[17,22,72]. The Chorrillo Formation was initially named as 'dinosaur-bearing strata' because of the abundance of fossil vertebrate remains that were found on these deposits[14–16,19–21,80,81]. However, studies carried out in the last years indicate that the Chorrillo Formation was not only rich in vertebrates but also in invertebrates, plants, and palynomorphs[14,17,22,23].

The arthropod specimens were obtained from a stratum located 215 m from the base of the Chorrillo Formation, which was named as Megafloral level 1[22], cropping out in uppermost part of the gullies of the La Anita farm (50°30'49.3"S 72°33' 35.9"W) (Fig. 7A). The insect-bearing strata is ~2.5 m thick and

consists of a dark gray to black colored, organic-rich, mudstone bed with diffuse parallel lamination[17] (Fig. 8). This bed accumulated in a waterlogged, low energy, environment as a swamp/pond, located in the distal areas of the low-gradient floodplain of the fluvial system, as the product of fluvial overbank processes[17,22]. This swamp/pond deposits are vertically related to Histosol-like paleosols[72]. The high content of organic matter in this bed suggests that vegetation grew in proximity to the swamp/pond favoring the development of the Histosol-like paleosol. The poorly decomposed state of the organic matter indicates anoxic and reducing conditions in the bottom water of the swamp/pond[72,82,83], which may also favor the exceptional preservation of non-mineralized organisms such as the arthropods here studied.

Detailed paleobotanical studies[22] from the same levels that contain the paleoentomofauna here presented have been recently published, allowing to characterize the floristic content accompanying the arthropods. Terrestrial elements of the assemblage include an array of spore producing taxa (mosses, lycopsids, and ferns), pollen grains referable to conifers of the families Araucariaceae, Podocarpaceae and Hirmeriellaceae ( = Cheirolepidiaceae), and dicot and monocot angiosperms[22]. While these floristic elements can have some degree of allochtony, the assemblage is also composed of aquatic plants. Recognized aquatic elements (e.g., Nymphaeaceae, Marsileaceae, Salviniaceae, freshwater algae) indicate the presence of a freshwater community that inhabited a low energy, paludal or marginal, mesotrophic environment. These taxa can be segregated in three functional groups, with the recognition of free floating macrophytes (*Azolla* spp.), floating leaved macrophytes (*Nuphar*?: Nymphaeaceae), and emergent species (*Gabonisporis vigorouxii*: Marsileaceae), distributed along a depth gradient[22]. This floral community probably acted as a food source and refugia for some of the aquatic insects here described.

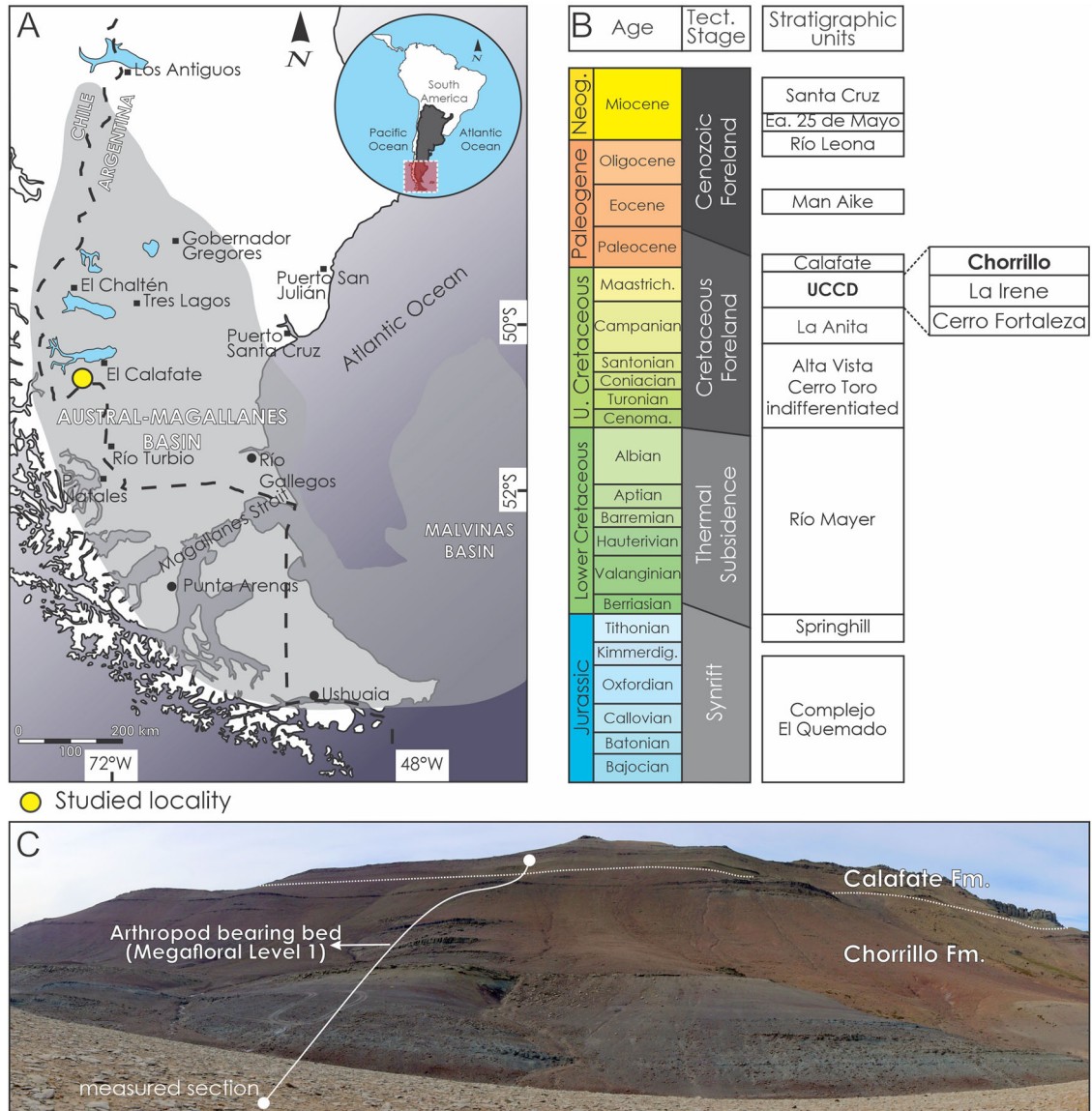

**Fig. 7 Location map indicating provenance of arthropod fossil remains. A** Location map of the Austral-Magallanes basin. **B** Stratigraphy of the sedimentary infill of the basin related with the three main tectonic stages. **C** Outcrop photopanel of the Chorrillo Formation at the study area showing the path of the measured sedimentary section and location of the arthropod-bearing bed.

**Sample processing**. The discovery of chitinous remains was fortuitous, after preparing fragments of rocks with the usual techniques employed for palynology. Rock samples were obtained from a stratigraphic level named Megafloral Level 1[22]. Superficial rock and debris was removed prior to collecting the samples, to avoid incorporating meteorized rock and/or foraneous organic content. Rock samples were treated using standard palynological techniques for extraction and concentration of palynomorphs[84] with a sequential combination of HCl (40%) and HF (70%). To increase the possibility of recovering large specimens, rocks were not crushed before the acid treatment. The remaining organic residue was sieved using 80 μm mesh to obtain the larger fractions, to recover organic macrofossils. The fractions of the residue with particles smaller than 80 μm were also revised. The organic residue was mounted and fixed in glass slides using glycerine jelly, as the original intention was to use them for a palynological study. Thus, the arthropod remains were serendipitously discovered while analyzing the slides searching for plant-derived remains. The fixed nature of the resulting microscope slides

precludes the recovery or removal of any particular specimen (e.g., each chironomid remain), for example to rotate it, without destroying the microscope slide sample. This fact also forbids the use of a scanning electronic microscope in the study, as the samples are encased between two fixed glasses. As a result, the illustration and description of the arthropod remains were carried on using a transmitted light microscope exclusively.

Coincidence in color (i.e., yellow to pale brown) and preservation quality among arthropod remains and other preserved elements (spores, pollen, plant cuticular remains, and other non-pollen palynomorphs[22]), fragmentation state, and diversity suggest the insect content cannot be interpreted as contamination of samples.

**Photography**. Specimens were studied using a Nikon SMZ800 binocular microscope, photographed using an attached camera (Nikon Coolpix 995) or a Canon Powershot SX540 HS digital camera from Museo Argentino de Ciencias Naturales "Bernardino Rivadavia", Buenos Aires. For clarity's sake, the CombineZP

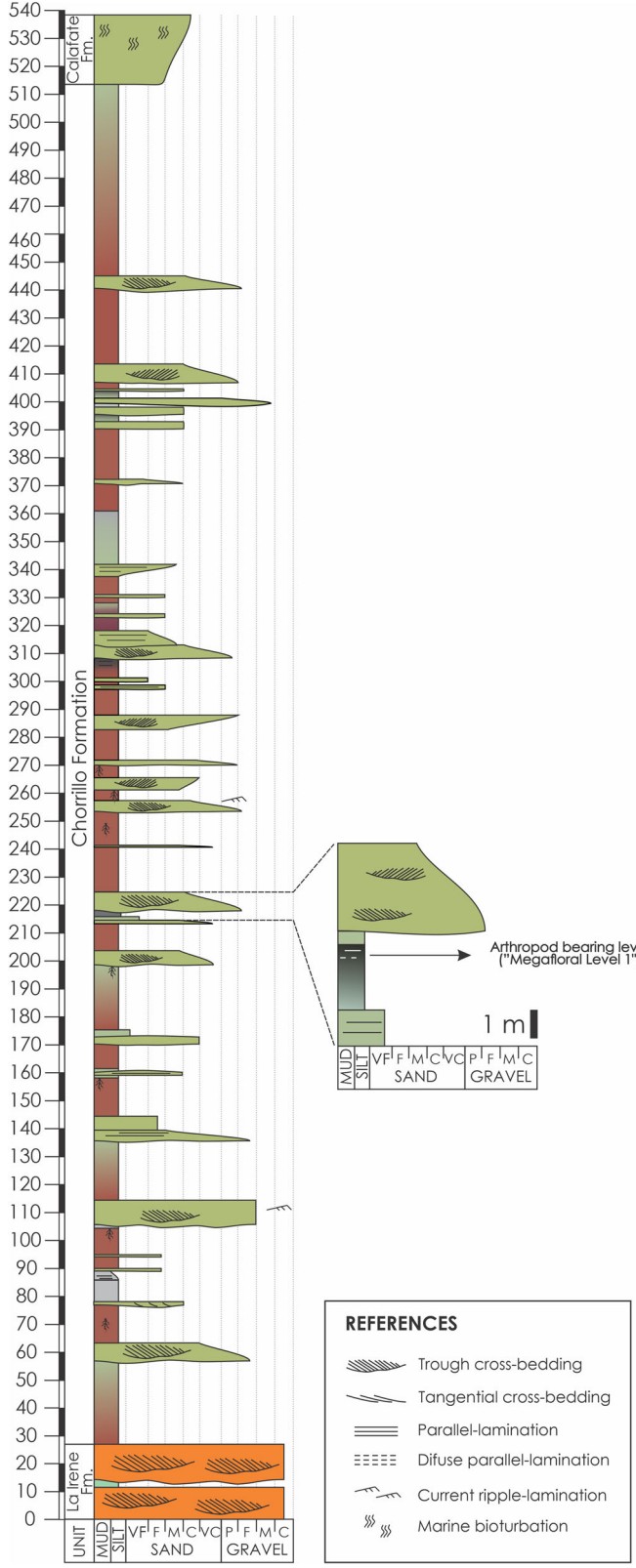

**Fig. 8 Estratigraphic column indicating provenance of arthropod fossil remains.** Detail of the sedimentary measured section of the Chorrillo Formation showing the main lithological and sedimentological aspects of the unit, the fossil content, and the detail of the arthropod-bearing bed (Megafloral level 1).

software (v.1.0) by Alan Hadley was used to produce photographic stackings of the identified elements, as its size and volume resulted in many focal planes.

**Reporting summary**. Further information on research design is available in the Nature Portfolio Reporting Summary linked to this article.

### Data availability
Arthropod materials are represented by approximately 30 specimens, deposited in the Museo Regional Padre Jesús Molina (Río Gallegos, Santa Cruz Province), under MPM-Pal 21835 (Palynological slides) acronyms. Respective collection numbers are as follows: MPM-Pal 21835-24: M31/0 (Orthocladiinae); MPM-Pal 21835-3: P46/4 (Orthocladiinae); MPM-Pal 21835-25: V26/3 (Orthocladiinae); MPM-Pal 21835-9: E34/4 (Diamesinae); MPM-Pal 21835-19: V34/4 (Diamesinae); MPM-Pal 21835-2:Y40/3 (Tanypodinae); MPM-Pal 21835-9: Q58/3 (Chironomidae indet.); MPM-Pal 21835-9:W37/1 (Ephemeroptera indet.); MPM-Pal 21835-10: X30/2 (Lepidoptera indet.); MPM-Pal 21835-7:S27/2 (Lepidoptera indet.); MPM-Pal 21835-3: W34/1 (Coelolepida); MPM-Pal 21835-23: W55/0 (Coelolepida); MPM-Pal 21835-22:R52/2 (Coelolepida); MPM-Pal 21835-20:Z47/3 (Coelolepida); MPM-Pal 21835-22: G55/0 (Coelolepida); MPM-Pal 2183-21: E42/3 (Coelolepida); MPM-Pal 21835-7:G42/4 (Arthropoda indet.); MPM-Pal 21835-23:H55/0 (Arthropoda indet.); MPM-Pal 21835-20:U27/0 (Arthropoda indet.); MPM-Pal 21835-4 (Arthropoda indet.); MPM-Pal 21835-22:W55/2 (Arthropoda indet.); MPM-Pal 21835-7:K21/2 (Arthropoda indet.); MPM-Pal 21835-23:L36/0 (Arthropoda indet.); MPM-Pal 21835-4:R46/0 (Arthropoda indet.); MPM-Pal 21835-23:W46/0 (Arthropoda indet.).

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

## Acknowledgements

The present paper is the result of the first Argentine-Japanese exploration, carried out in March 2020. We thank Coleman Burke (New York), for his encouragement and financial assistance to carry on first field explorations to La Anita farm. Special thanks to Dr. Y. Harashi, former General Director of the National Museum of Nature and Science, Japan. Thanks are extended to editor George Inglis and three anonymous reviewers for their critical comments, which greatly improved the final version of the manuscript. Special thanks to Federico Braun for allowing access to his property. Facundo Echeverría and his wife Daphne Fraser (La Anita farm) offered their valuable geographic knowledge of these territories, allowing us an easy access to fossil sites with our 4x4 vehicles. Oscar Canto and Carla Almazán (Secretaría de Cultura) for supporting our projects and explorations in Santa Cruz Province. We thank Jorge Genise for reviewing an early version of this manuscript.

## Author contributions

E.I.V. and V.P.L. collected and processed the samples, and discovered the insect specimens. D.M.P. carried out the stratigraphic work. M.D.M., J.M., L.M.S., O.F.G., E.I.V., and F.L.A. carried out systematics and descriptions of the specimens. E.I.V., F.L.A., and F.E.N. conceptualized the study. F.E.N., M.M., and T.T. acquired fundings. VPL, EIV, D.M.P., and F.L.A. made the figures. All authors contributed to the writing and general revision of the manuscript.

## Competing interests

The authors declare no competing interests.
