## [Peer Review File · Communications Biology]

Reviewers' comments:

Reviewer #1 (Remarks to the Author):

This is a well-written paper reporting the first discovery of Maastrichtian insect assemblage from southern Patagonia. I am very glad to know latest Cretaceous insects are found from the Southern Hemisphere. I have a few questions and suggestions and I suggest acceptance after moderate revision.

Fig. 1n-w: Why these microfossils are insects, not other arthropods? How can you distinguish them? Some fossils like fig. 1o looks strange. Maybe the authors can use SEM to take some clearer images. The main text is short, with only one figure. I suggest to move the SUPPLEMENTARY INFORMATION to the main text.

Some minor suggestions are as follows:

Title: What is "arthropod dynamics" mean? This phrase does not appear anywhere else in the whole paper.

Line 92: where is fig.2? Please check.

Lines 106-109: "The presence of Orthoclaadiinae and Diamesinae" suggests low productive and vegetated, shallow waters, but "the presence of Diamesinae" indicates running waters? Looks contradictory.

Lines 114-115: "In contrast, less than 10% of the specimens collected in Cretaceous amber inclusions correspond to Chironomidae". Indeed, but it can also be explained by chironomid larvae are aquatic and not so easily caught by amber. So maybe better to compare with lithological insect biota.

Reviewer #2 (Remarks to the Author):

Vera et al. documented an interesting insect assemblage between Cretaceous and Paleogene age. This paper would be a valuable contribution to the research of fossil insects.

1. All the species were preserved as fragments, and it is different to assign them into definite taxa in family level even order level within insects. More photos under different views about these specimens would be useful for identification. Authors should offer more figures in Supplementary Information and mark the key structures for further classification, which is the foundation for the discussion of this paper.

2. Scales were also found on the wings of Cretaceous caddisflies, see wang et al. 2022 in Current Biology, should be compared.

3. I suggest moving the "SI figure 1" of "Paleoforistic setting" to the main text.

4. The format of the references is problematic. Please double check the citation format of the references and modify them.

Others.

Line 29: do not repeat the word "southern Patagonia", which is mentioned in title.

Line 10, line 11 (SI): change "channel-shape and lobe-shape" to "channel-shaped and lobe-shaped".

Line 81 (SI): should be "double median teeth".

Line 113 (SI): should be "compound eyes"

Reviewer #3 (Remarks to the Author):

The authors present a clear and well written manuscript on the discovery of insect remains in Cretaceous sediments from Argentina collected for palynological analysis. Thus the samples were observed between blade and bladelets, they therefore constitute elements of very small size. The period is indeed very interesting for insects because we have very few deposits from the end of Cretaceous , useful for assessing the impact of the K/T crisis.

The authors do not specify the palynological methods used, which could have an influence on the data (size bias for example).

The diversity of Lepidoptera is suggested by the diversity of scales while it seems that several types of scales can be present on the same taxa.

The taxonomic assignments are given on the basis of comparison of anatomical elements which are not figured (with their current equivalents).

There is a difference between the announcement effect of this deposit and the reality of the insects found, few in number and not very infirming in the state of what is exposed.

This seems a bit hasty publication of preliminary results.

The authors do not specify enough the sedimentary context and the methods for example not to have contamination (scales of Lepidoptera for example).

Otherwise the importance of this type of deposit is proven and we would like to know if there is a potential to complete the sampling.

Buenos Aires, August 14th, 2023

Dear Dr. George Inglis,

I am sending back to you the revised version of our ms entitled "**Maastrichtian insect assemblage from Patagonia sheds light on arthropod diversity previous to the K/Pg event**". The new version has been considerably improved by following most of the observations and suggestions expressed by the three reviewers. We deeply thanks for their appropriate comments.

Response to Reviewer #1:

"This is a well-written paper reporting the first discovery of Maastrichtian insect assemblage from southern Patagonia. I am very glad to know latest Cretaceous insects are found from the Southern Hemisphere. I have a few questions and suggestions and I suggest acceptance after moderate revision."

RESPONSE: thanks for these encouraging comments.

"Fig. 1n-w: Why these microfossils are insects, not other arthropods? How can you distinguish them? Some fossils like fig. 1o looks strange."

RESPONSE: we have saved this mistake in the new version, by now referring to as "arthropod/s", rather than "insects".

"Maybe the authors can use SEM to take some clearer images."

RESPONSE: as explained above, this is impossible to do with current samples (unfortunately), because they are embedded in jelly and glass.

"The main text is short, with only one figure. I suggest to move the SUPPLEMENTARY INFORMATION to the main text."

RESPONSE: following this recommendation, we have now incorporated into the main text, all the taxonomic information originally expressed in the SI, and expanded the number of figures.

"Some minor suggestions are as follows:

Title: What is "arthropod dynamics" mean? This phrase does not appear anywhere else in the whole paper."

RESPONSE: following Reviewer#1, we have replaced "dynamics" for "diversity" in the title.

"Line 92: where is fig.2? Please check."

RESPONSE: this was a typographic error, as no figure 2 was present in the originally submitted manuscript.

"Lines 106-109: "The presence of Orthoclaadiinae and Diamesinae" suggests low productive and vegetated, shallow waters, but "the presence of Diamesinae" indicates running waters? Looks contradictory."

RESPONSE: It is not contradictory. In the manuscript we suggest a shallow water lacustrine environment; however it is common to find running-water elements in lacustrine deposits as they can be transported to the lake/pond via runoff through erosion, precipitation, etc. Therefore, the presence of both, lentic and lotic elements (such as Diamesinae) in the Chorrillo Formation is indicative of a lacustrine environment with influence of running waters.

“Lines 114-115: “In contrast, less than 10% of the specimens collected in Cretaceous amber inclusions correspond to Chironomidae”. Indeed, but it can also be explained by chironomid larvae are aquatic and not so easily caught by amber. So maybe better to compare with lithological insect biota.”

RESPONSE: Chironomids are almost absent in Cretaceous lithic deposits. Thus, amber deposits represent (for the time being) one of the most reliable sources about Cretaceous chironomids. We agree with the reviewer in the fact that this type of preservation (amber) will result in an underrepresentation of chironomid larvae (as well as other aquatic animals). Thus, we incorporated a sentence indicating this fact.

Response to Reviewer #2

“Vera et al. documented an interesting insect assemblage between Cretaceous and Paleogene age. This paper would be a valuable contribution to the research of fossil insects.”

RESPONSE: thanks for these encouraging comments.

“1. All the species were preserved as fragments, and it is different (*sic*; NOTE TO THE EDITOR: we believe that the right word is “difficult”) to assign them into definite taxa in family level even order level within insects. More photos under different views about these specimens would be useful for identification. Authors should offer more figures in Supplementary Information and mark the key structures for further classification, which is the foundation for the discussion of this paper.”

RESPONSE: Although we agree with the reviewer regarding incorporation of additional images, we have to emphasize that the restrictions imposed by the way specimens are mounted forbid us to produce substantially new images (it is impossible to take lateral or back images, unfortunately). As explained above, the specimens were identified in permanent pollen slides, making impossible to remove them from the mounting media (otherwise, the risk of destruction of the materials is high).

However, let us note that the images we provided in both original and current submissions, were produced with photographs taken at different depth levels, thus producing a composite photo holding (as much as possible) morphological structures with taxonomic information. Most of the images of the illustrated specimens consist of photographic stackings made with the Combine ZP software, using photographs taken at different focal planes.

Besides, we entirely disagree with Reviewer#2 when he says that “it is difficult to assign them into definite taxa in order or family levels”,

mistrusting of our taxonomic work. We are confident with the identifications offered in the text, based on comparisons in museum collections, consult of bibliography, and familiarity with insect anatomy and systematics. Even the images provided in the manuscript (with the limitations already explained) show clearly enough features supporting taxonomic referrals to different groups (i.e., Fam Chironomidae, Ephemeroptera, Lepidoptera).

“2. Scales were also found on the wings of Cretaceous caddisflies, see Wang et al. 2022 in Current Biology, should be compared.”

RESPONSE: we have incorporated this information into the present version.

“3. I suggest moving the “SI figure 1” of “Paleoforistic setting” to the main text.”

RESPONSE: done

“4. The format of the references is problematic. Please double check the citation format of the references and modify them.”

RESPONSE: done

“Line 29: do not repeat the word “southern Patagonia”, which is mentioned in title.”

RESPONSE: done

“Line 10, line 11 (SI): change “channel-shape and lobe-shape” to “channel-shaped and lobe-shaped”.

RESPONSE: done

“Line 81 (SI): should be “double median teeth”.

RESPONSE: done

“Line 113 (SI): should be “compound eyes”

RESPONSE: done

Response to Reviewer #3

“The authors present a clear and well written manuscript on the discovery of insect remains in Cretaceous sediments from Argentina collected for palynological analysis. Thus the samples were observed between blade and bladelets, they therefore constitute elements of very small size.”

RESPONSE: thanks for these encouraging comments.

“The period is indeed very interesting for insects because we have very few deposits

from the end of Cretaceous, useful for assessing the impact of the K/T crisis.”

“The authors do not specify the palynological methods used, which could have an influence on the data (size bias for example).”

RESPONSE: We expanded the methodological section, including information of the treatment carried out to obtain the organic remains from the rock sample, sieving and study using light microscopy

“The diversity of Lepidoptera is suggested by the diversity of scales while it seems that several types of scales can be present on the same taxa.”

RESPONSE: although this observation is logical (and we have modified the text to express this problem), we still believe that the variety of scales surpass the variety in shape of a single individual/species. In sum, we follow recommendation of Reviewer#3 in being more cautious. Following reviewer’s observation, we have also included a brief sentence indicating the differences between trichopteran and lepidopteran scales, based on Wang et al. (2022) and Van Eldijk et al., (2018).

“The taxonomic assignments are given on the basis of comparison of anatomical elements which are not figured (with their current equivalents).”

RESPONSE: we have modified the figures according to this recommendation.

“There is a difference between the announcement effect of this deposit and the reality of the insects found, few in number and not very infirming in the state of what is exposed.”

RESPONSE: Let us disagree with this comment. The fossil record of continental arthropods from the latest Cretaceous is poor, and the one of Maastrichtian age almost non-existent, thus our contribution substantially expands the knowledge of the continental aquatic biota.

“This seems a bit hasty publication of preliminary results.”

RESPONSE: We don’t agree with this comment. The observations, comparisons and interpretations we offer on the morphology, taxonomy, and paleoecology of these --for the first-time documented Maastrichtian insects-- are sound and novel. As already expressed, limitations are imposed for the way the specimens have been prepared.

“The authors do not specify enough the sedimentary context and the methods for example not to have contamination (scales of Lepidoptera for example).”

RESPONSE: We agree with the reviewer on this comment. We have expanded the “Geological setting” which is now named “Geological and paleoenvironmental context” in the revised version of the manuscript. In the fresh version of the manuscript, we have expanded the sedimentological and environmental implications of the arthropod bearing strata and the possible controls on the exceptional preservation of these remains.

We also expanded the methodological section to clarify how samples were obtained and processed, and why we interpret the arthropod remains as fossils contained in the strata, and not contamination.

“Otherwise the importance of this type of deposit is proven and we would like to know if there is a potential to complete the sampling.”

RESPONSE: as the Reviewer#3 say, this layer of the Chorrillo Formation has proved to be an interesting source of novel information on latest Cretaceous insects, which we are exploring just at the beginning

Sincerely yours,
Dr. Fernando E. Novas

REVIEWERS' COMMENTS:

Reviewer #1 (Remarks to the Author):

The authors have successfully addressed all concerns, improving the manuscript with their edits. This will be a fine contribution as it stands. I recommend acceptance in this form.

Reviewer #2 (Remarks to the Author):

The authors have well addressed all the previous comments. I think this manuscript is in good shape now. Congratulations!